# An Overview of Malaria Transmission Mechanisms, Control, and Modeling

**DOI:** 10.3390/medsci11010003

**Published:** 2022-12-23

**Authors:** Merveille Koissi Savi

**Affiliations:** 1Center for Development Research ZEF, University of Bonn, Genscherallee 3, 53113 Bonn, Germany; merveillekoissi.savi@gmail.com or mksavi@hsph.harvard.edu; 2T.H. Chan Harvard School of Public Health, Boston, MA 02115, USA

**Keywords:** complexity, policy-recommendation, mathematical models, malaria, mosquito

## Abstract

In sub-Saharan Africa, malaria is a leading cause of mortality and morbidity. As a result of the interplay between many factors, the control of this disease can be challenging. However, few studies have demonstrated malaria’s complexity, control, and modeling although this perspective could lead to effective policy recommendations. This paper aims to be a didactic material providing the reader with an overview of malaria. More importantly, using a system approach lens, we intend to highlight the debated topics and the multifaceted thematic aspects of malaria transmission mechanisms, while showing the control approaches used as well as the model supporting the dynamics of malaria. As there is a large amount of information on each subject, we have attempted to provide a basic understanding of malaria that needs to be further developed. Nevertheless, this study illustrates the importance of using a multidisciplinary approach to designing next-generation malaria control policies.

## 1. Introduction

Approximately 2000 children die daily from malaria, with 90% of the victims located in sub-Saharan Africa (SSA) [1,2]. Transmission and control of malaria are mediated by complex interactions and feedback loops among humans, mosquitoes, parasites, their environments, healthcare systems, and policy implementation at a given period of time [3]. Therefore, malaria transmission can be described as complex, nonlinear, and dynamic. The endemism of malaria in West Africa, for example, has huge economic impacts [2,4]. In Ghana, the cost of treating a single episode of malaria can reach 34% of a household’s yearly income [5]. As a result of infected people being unable to produce wealth, and as wealth is low, the risk of exposure to malaria increases. There is a negative relationship between national economic growth and malaria control expenditures [2,6]. As a result, malaria can be considered both as a cause of poverty and as a consequence of it [7].

Furthermore, past interventions have been ineffective as they were unable to account for the nonlinear nature of malaria infection. Particularly, the unexpected consequences of interventions such as mosquito resistance to chemical compounds and parasite resistance to drugs led to their abandonment. This was the case with the eradication control program that was abandoned in 1969 and the Global Malaria Control Strategy in 1992 [8]. Scientists have widely acknowledged the complexity of malaria, but a depiction of this complexity is weakly documented, and further limits the scope of malaria control.

This short review embracing several topics or thematic areas provides the reader with a piece of basic knowledge of malaria transmission, control, and modeling while illustrating its complexity. Therefore, the manuscript intends to be a didactical material, providing a brief overview of the disease, reducing information asymmetry, while arousing the curiosity of readers who can refer to more specific literature.

## 2. Materials and Methods

We conducted an unsystematic narrative review of online literature to gain a deeper insight into malaria transmission mechanism, control strategy, and modeling approaches. Thus, we adopted the following approach (Figure 1).

### 2.1. Inclusion and Exclusion Criteria for Paper Eligibility

Our search for peer-reviewed papers on malaria in the online scientific database was restricted to the period from 1950 to 2020. The rationale for this time window lies in the fact that the first malaria model was developed in the 1950s and since then, most of the recommendations for malaria control are supported by quantitative analysis. Besides, we excluded all duplicated letters, narrative reviews, case reports, and commentary. Given the tremendous research done on malaria, we limited the scope of the study to only peer-reviewed published papers in either French or English. This was to avoid evidence that may be challenging to replicate and biased translation. Within the topic of interest, we exclude all case reports which happen in non-endemic areas.

### 2.2. Search Strategy

We searched three databases PUBMED, Google Scholar, and MEDLINE for relevant literature on malaria from 1950 to 2020.

### 2.3. Screening of Studies and Evidence Extraction

Our screening process focuses on the title of the study published in peer-reviewed journals. Our next step was to review the papers’ abstracts and remove the studies that did not fit under the umbrella of the keywords.

### 2.4. Analysis and Gaps in the Analysis

To get a comprehensive understanding of malaria, we categorized the list of 77 eligible peer-reviewed papers into topics of interest, such as mechanisms, prevention and treatment, mathematical modeling, and perspectives on modeling. The topical organization of the collected papers allowed us to get a broad perspective of the state of the art.

## 3. Results

We found 2,150,000 studies after the search. A total of 150,000 studies were obtained after removing duplicates. We included only 300 studies after screening both titles and abstracts. Furthermore, we removed 50 case reports, 146 studies in malaria non-endemic areas, and 16 reports including 10 thesis and 6 NGO reports.

Using the remaining 77 eligible studies, we provided an overview of each topic and complemented our narrative with WHO, CDC reports, and 10 complimentary readings. There were 23 studies supporting malaria transmission, 19 on prevention and treatment, 23 on mathematical modeling, and 12 on future research on malaria modeling (Figure 2).

### 3.1. Malaria Transmission Mechanism

Malaria infection is caused by interactions between humans, mosquitoes, and the pathogen (Figure 3). Female mosquitoes change during the egg-laying period when they experience a deficiency in protein and plant sucrose necessary for egg maturation. Consequently, they blood-feed on humans and can infect them after effective biting where infectious protozoans (of the genus *Plasmodium*) are injected into the human bloodstream [9,10].

Female mosquitoes most found in West Africa belong to the *Anopheles* Giles genus [11]. *Anopheles gambiae* (*sensu stricto*) prefers to use human blood (anthropophilic) to complete its gonotrophic cycle, bites primarily indoors (endophagic), and rests indoors (endophilic). As far as its feeding habits are concerned, *An*. *gambiae* is the major malaria vector in West Africa [12]. In addition, there are other vectors, such as *An. arabiensis*, *An. funestus* and *An. melas* able to carry *P. falciparum*. Similar to *An. gambiae*, *An. funestus* is anthropophilic, endophagic, and endophilic [4,13]. On the other hand, *An. arabiensis* is zoophilic (prefers blood-feeding animals), exophilic (bites outdoors), and exophagic (rests outdoors) [11]. *An. melas* is also anthropophagic, exophagic, and exophilic [12]. Therefore, *P. falciparum* adapts well to Anopheles species, increasing malarial vulnerability in the host. The wide range of vector conditions, feeding habits, and geographical heterogeneity in malaria transmission make malaria modeling even more challenging [14].

A mosquito’s ability to adapt to environmental conditions such as humidity and temperature, is influenced by its sensitivity to those conditions. This adaptation can take the form of a change in the life cycle length, or the biting efficiency [15,16,17,18,19]. As a long-term study carried out in Kenya (1999–2010) has shown, vectors’ feeding habits and effectiveness can change over time [20].

The transmission intensity may differ significantly across host populations because vectors’ attraction to the host varies [21]. This variation is elicited by the difference in human bodies’ microflora and their reaction to mosquito bites [22]. For instance, an infected host is more attractive to other mosquitoes than a non-infected one [23]. Likewise, mosquitoes prefer the scent of children to that of the adults [21,24].

In SSA, infectious protozoans are, *Plasmodium vivax, P. ovale, P. malariae, P. knowlesi,* and *P. falciparum* where *P. falciparum* accounts for 95% of malaria cases [25,26]. *P. falciparum*’s complete life cycle consists of an incubation period of 10 days in the vector and a period of 7 to 20 days in the human host [27]. When *P. falciparum* protozoans are injected into a host’s bloodstream via the vector saliva, the protozoan completes its life cycle. After entering the host bloodstream, the sporozoites (asexual protozoans) move into the liver, where they penetrate the hepatocytes (liver cells). In the host hepatocytes, the parasite divides asexually several times, a process known as schizogony [28]. Eventually, the hepatocytes break, releasing merozoites which infest red blood cells in the host bloodstream. The parasites can diverge into gametocytes after repeated divisions of the red blood cells [29]. Malaria can cause symptoms such as fever, chills, and sweat before and during the differentiation of *P. falciparum* into sexual stages. Even so, some hosts may not exhibit clinical symptoms and are considered asymptomatic [30]. Whenever mosquitoes bite infected people (symptomatic and asymptomatic), they primarily suck out the gametocytes [29]. Detecting asymptomatic individuals in endemic areas can be challenging, and mass testing may not always be cost-effective in detecting such individuals [31].

### 3.2. Malaria Prevention and Treatment

Due to recursive interactions between humans, mosquitoes, and parasites that cause malaria transmission, existing control methods might not be effective in the sense that they target either mosquitoes or parasites. Mosquitoes have been controlled mainly using the obliteration of larvae and adult stages while reducing the entomological inoculation rate. These controls include physical, biological, or chemical changes to the vector’s environment. As part of the physical modification, breeding sites are removed, and the sources of larvae are managed using drainage and weeding [32]. However, this method requires a constant working force needed for weeding and drain cleaning which led SSA governments to abandon it although it is among the most effective [33]. Chemical modification of larval environments includes the application of larvicides and insecticides. The downside of this method is that it is also costly, and mosquitoes often become resistant to insecticides [34]. Natural enemies of mosquito larvae, such as larvivorous fish and bacteria (e.g., *Bacillus thuringiensis*), are used in the biological method. Nevertheless, this process is hindered by the temporary persistence of natural enemies [32].

The Global Malaria Action Plan (GMAP) and the World Health Organization (WHO) recommended the use of indoor residual spray (IRS) and insecticide-treated bed nets (ITNs) for controlling mosquito adult populations in the 2010s across Africa [35]. The main chemical component of ITNs and IRS is synthetic pyrethroid, a lethal compound, that repels mosquitoes, remains in the environment, and is harmless to mammals. From 2005 to 2010, malaria cases in Ghana declined by 41% due to the successful use of ITNs and IRS [36]. The residual effects of synthetic pyrethroids in the environment resulting from IRS, ITNs, and agricultural pest management force mosquitoes to develop coping mechanisms [37]. These include indoor and outdoor spray avoidance mechanisms, earlier biting, and insecticide resistance [38].

The control of malaria parasites is achieved through three lines of defense: antimalarial drugs, seasonal prevention, and vaccination. The first line of the curative measures adopted by GMAP is the Artemisinin-based Combination Therapies (ACT). ACTs are highly potent against *P. falciparum* with a higher clearance rate and symptoms resorption than other curative therapies [39]. Nevertheless, *P. falciparum* has progressively developed ACT resistance [40]. Malaria parasite resistance in SSA is exacerbated by using counterfeit malaria drugs [41,42]. Another preventive measure is intermittent prevention, which involves administering single-dose malaria therapy to pregnant women [43]. The treatment includes the combination of primaquine, sulfadoxine-pyrimethamine, amodiaquine, methylene blue, and dihydroartemisinin-piperaquine which have been proven to be able to prevent the transmission of *P. falciparum* [44]. A seasonal vaccine “RTS, S/AS01” is proven effective for children under 5 years old for a short period with an efficacy rate of 82% [45,46]. The authors demonstrated that RTS, S/AS01 induces an immune response to the proteins of infected sporozoites. Vaccine components protect against clinical malaria. The effectiveness of its protective action can wane over time and with advancing age. Particularly, children between the ages of 6 to 12 weeks benefit from the vaccine more than children between the ages of 5 to 17 months.

A less explored line of malaria treatment consists of phytotherapy. For example, a study on phytochemicals identified more than 20 local plants and herbs in central Africa that can heal and prevent malaria transmission with almost zero risks of *P. falciparum* developing resistance against these natural compounds [47,48,49]. According to the authors, the synergistic interplay between a wide range of chemical substances in the plants enables treating malaria holistically while preventing the development of resistance [48,50].

### 3.3. Mathematical Modeling of Malaria

Mathematical models developed to gain insight into disease transmission have influenced past and present interventions to prevent or treat diseases. Mathematical modeling of malaria transmission began in 1911 with the Susceptible-Infectious-Removed model (SIR), which compartmentalized the population of hosts and vectors into three groups [51,52]. The compartments were denoted susceptible (S), referring to the population likely to become infected, infectious (I), composed of the fraction of the infected population, and removed/ recovered (R), accounting for the fraction of the population that either died or healed from the disease. Several assumptions were assumed in the SIR model, including a closed and finite population, a homogeneous mosquito bite rate, and a well-mixed population. Although the Ross model (SIR) was unable to adjust to the new incidence data due to its limited predictability, SIR provides insights into the intricate relationship between infected hosts and mosquito densities. Because Ross’s model was theoretical and simulation-based, it failed to document proof for campaigns for mosquito eradication.

Subsequently, George Macdonald complemented the SIR model [51] by feeding the model with real data and embedding an additional compartment for the latency period between the mosquito bite and the onset of symptoms denoted Exposed (E). Additionally, Macdonald’s study argued for eradicating mosquitoes through a massive campaign based on the theory of superinfection [53]. Consequently, between 1955 and 1969, WHO implemented widespread and rigorous mosquito eradication campaigns as part of the Global Malaria Eradication Program (GMEP). In these campaigns, the pesticide Dichlorodiphenyltrichloroethane (DDT) was successfully used to control malaria in major European and American countries [54]. The GMEP, however, failed to eradicate malaria worldwide due to mosquito resistance [55]. Apart from mosquito insecticide resistance, the absence of basic healthcare services, high malaria transmission intensity, and other socio-ecological factors hindered the success of eradication campaigns in the SSA [56].

The mathematical epidemiology of malaria has evolved steadily in recent decades from “toy models” (which were not realistic but captured the key features of the disease) to “high-level models” (which are precise and sacrifice generality) [57]. As such, complex models display common features such as the interaction between many components [58]. For example, an earlier partial differential model split the infected population into age-dependent infections in the SIR framework [59].

An important metric that summarizes the transmission of a disease is the reproductive number (R_0_). R_0_ represents the expected number of infected human hosts after effective mosquito bites in a fully susceptible population [60,61,62]. Thus, R_0_ provides a measurement for the intensity of transmission and contributes to the definition of disease-endemic areas when it is greater than one (R_0_ > 1) [63]. Heterogeneity was therefore incorporated in populations for the computation of R_0_. These include, for example, the age structure of the host [64], migration of the vectors and hosts [65], host beliefs and practices [66], and host income classes [67,68]. R_0_ also varies with the degree of complexity introduced into the compartmentalization of the population [66]. The more the modeling framework embodies the heterogeneity in the host population, the closer to real-life transmission dynamics the value obtained for R_0_ becomes [63,68,69,70]. The pioneering models that provided insight into malaria dynamics and its controls were unrealistic for various reasons, and maximum control has hardly been achieved in [56]. Most specifically, a large R_0_ indicates a higher density of mosquitoes that reduces the effectiveness of ITN in a large and likely heterogeneous host population [63].

Many mathematical models were developed combining the compartments S, E, I, and R (e.g., SIR, SIS, SI, SIRS, SEIR, SEIRS, SEI) and included the heterogeneity of the population (meta-population modeling approaches) [69]. Most of these models do not capture the dynamics of subpopulations because they tend to assume large and homogeneous populations that do not take into account the existence and specific characteristics of subpopulations [71,72]. Another issue is that they fail to account for host behavior, which influences the disease’s dynamics and control [73,74]. Besides, compartmental models are either knowledge or data-driven (e.g., Ross model, Macdonald models). The models cannot address the complex interactions between determinants that guide the disease dynamics in both cases. For example, socioeconomic determinants interfere with the parameters generally considered in the calculation of R_0_ such as mortality, mobility, and birth rate [66]. Thus, models that capture, realistically, the transmission process of malaria are still missing.

### 3.4. Malaria Modeling: Present and Future

The ability to track epidemics became more accurate with the advancement in computing power and the availability of data. The deterministic framework uses in this context provides insight into disease dynamics, but usually fails to take uncertainties into account. Stochastic models, on the other hand, incorporate randomness that may occur during the epidemic and, most importantly, accounts for a large range of uncertainty that occurs during epidemics. For example, a stochastic model was used to predict the complex transmission pattern concerning environmental changes [75]. Like deterministic models, the stochastic models help to disclose interesting features of the transmission such as the impact of the network structure, and the properties of an outbreak most specifically get insights into the rate of the surge of vector-borne diseases in a new region in a context of climate change [76,77]. Furthermore, both stochastic and deterministic models are population-based and often fail to consider the feature of individuals which alternatively the individual-based models (IBMs) do [78]. IBMs can account for heterogeneity across individual agents and spatial gradients [79,80]. However, parametrizing them remains a daunting task, especially due to the lack and accuracy of data [81]. Beyond meta-population modeling (Ross-based models), within-host modeling opened new possibilities to address malaria infectiousness while examining the interplay between hosts’ immune systems and the dynamics of the parasite. However, modeling the superinfection of malaria while tracking each parasite strain and incorporating the diverse genetic makeup of the parasites remains extremely challenging, especially in terms of the model parameterization [82,83]. Most recently, the introduction of a new type of data such as human mobility collected from data providers (Facebook, Twitter, Google, etc.) refined the field of epidemiological surveillance. These data have been used to pinpoint malaria hotspots and predict their spread [84,85,86]. The use of fine-grained mobility data opens a new prospect of modeling and documenting evidence for policy recommendations; however, privacy concerns need to be addressed [87].

## 4. Discussion

Malaria remains a major health challenge in Sub-Saharan Africa, despite extensive research. Several factors contribute to malaria transmission, including socio-ecological factors and changes in the human host and vector behavior, which may be channeled by parasite genetic changes. Considering malaria’s complex nature, modeling it presents numerous challenges. As an example, it can be difficult to accurately parameterize a model with a compartment of asymptomatic individuals as there is a lack of reliable historical data. However, evidence suggests that malaria symptoms could progress rapidly from none to life-threatening while leading to a lifetime burden of cognitive impairments for some patients. It is thus necessary to develop a strategy that allows for the production at a low cost of reliable primary data on asymptomatic people with the capability of identifying infectious Plasmodium [88]. Additionally, the review highlighted that for each of the thematic areas discussed, there is a web of determinants that can be studied, using different perspectives including the use of a multidisciplinary lens to examine clinically relevant malaria issues.

A historical approach was used to illustrate the evolution of the malaria model from simplistic to more elaborate models. Moreover, this review outlines the challenges of modeling, where, for instance, introducing mobile data to a mathematical framework may pose a problem of re-identification. The number and scale of parameters involved in a complex modeling approach make model parameterization challenging.

Finally, this review demonstrates that malaria can efficiently be controlled by combining several strategies. Among them are constant awareness and education campaigns in communities at risk and among individuals whose resistance to non-pharmaceutical interventions (such as ITNs) can hinder malaria control efforts.

Given the wide range of thematic areas discussed, we acknowledge that we have only scratched the surface of the various cross-cutting topics. Consequently, we did not elaborate on the host immune response to malaria nor on technological advances in diagnosing and treating malaria since they are widely covered elsewhere [89,90,91,92,93,94]. Additionally, this paper faces two challenges, namely (1) a self-selection bias that might lead to a more prominent emphasis on some cross-cutting topics, and (2) not considering recently published papers [95]. Nonetheless, this review provides readers with an overview of malaria from a global perspective.

## Figures and Tables

**Figure 1 medsci-11-00003-f001:**
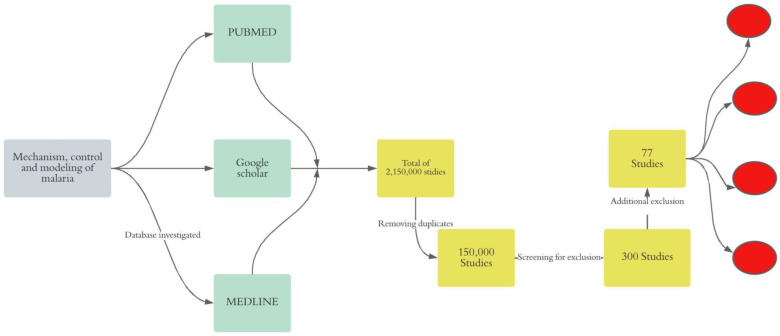
Synthesis of the method. In gray, the final objective of the review is shown, in green, the investigated databases are indicated, in yellow, the exclusion process is indicated, and in red, the clustering of the selected papers is shown.

**Figure 2 medsci-11-00003-f002:**
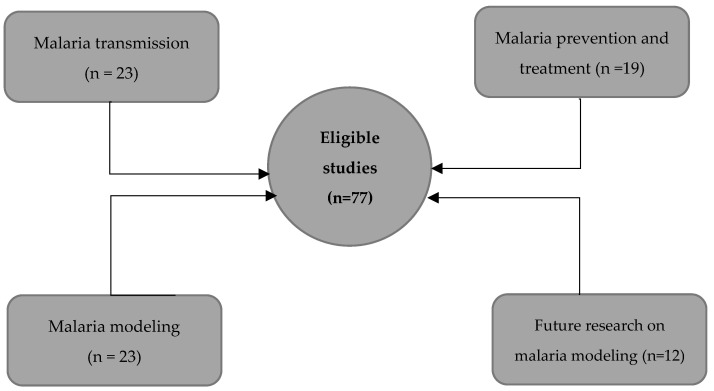
Overview of the thematic distribution of the selected papers.

**Figure 3 medsci-11-00003-f003:**
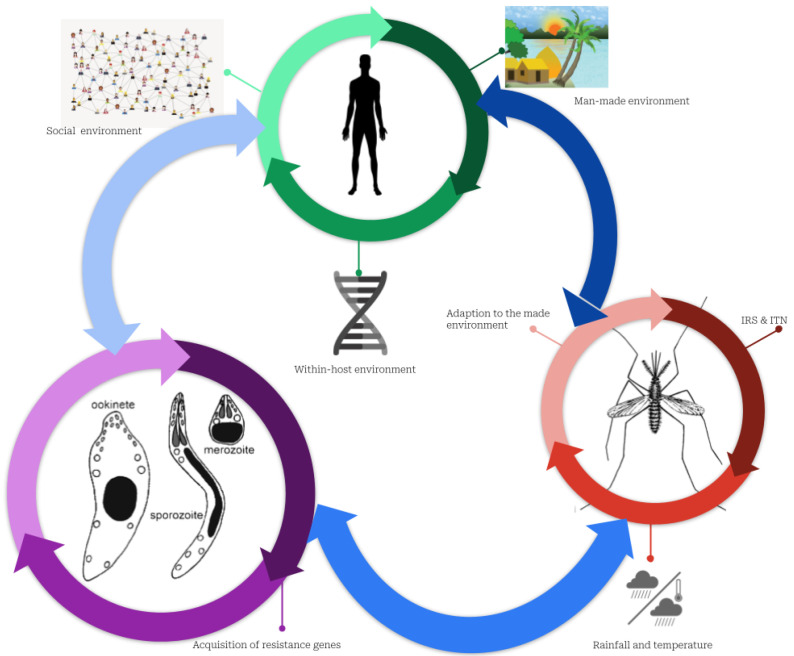
Complex interactions between humans, mosquitos, and parasites explaining the persistence of malaria. Plasmodium parasites, mosquitoes, and humans constitute three subsystems that interact with one another. The green cycle represents the human sub-system (SS). In addition to the micro-environment, which includes housing conditions, network connectivity, socioeconomic conditions, and the environment within the host (i.e., the individual’s genetic makeup), this SS is characterized by the social environment (social interaction, culture, and behavior regarding malaria risk-taking). The red indicates the SS of mosquitoes defined by the growth conditions favored by humans using or not either ITNs or IRS, and their adaptation to their living environment, including the harsh environmental conditions of cities, such as dirty water and weather conditions. In purple, Plasmodium parasite SS is characterized by interactions with the environment inside the host that may drive resistance to malaria drugs.

## Data Availability

All data are generated from the content of publications mentioned in this manuscript and they are equally available from the corresponding authors on reasonable request.

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
