# Peer review of "An Overview of Malaria Transmission Mechanisms, Control, and Modeling"

_medsci, 2022, doi:10.3390/medsci11010003_

Round 1

Reviewer 1 Report

The manuscript entitled "An overview of malaria transmission mechanisms, control, and modeling". Title, abstract and overall rationale of work is well written and property explain. However, there are still some major concerns, which needs to be addressed before publication..

1) Author must be include two more keywords such as Malaria, Mosquito or suitable keywords.

2) Author describe methodology and explain how they collected the articles from different sources. It is better a flowchart should be added to the article to show the clear and attractive methodology.

3) This is review article and author explain many things about the malaria transmission mechanisms, control, and modeling, unfortunately, author did not showing any figure related to mechanism, transmission control and other. I suggest author they must be incorporate the figure here.

4) Figure 2 is not attractive and low resolution. I suggest author they should increase the resolution and modified this representation.

5) Line number 88-90: author describe about the Plasmodium parasite transmission and they cite the references (9-10). These references are too old and it is better to incorporate new reference (doi: 10.1016/j.jare.2020.02.016) here.

6) Before used abbreviation author must be write first time in the manuscript after that they used. For example Anopheles gambiae. See the line number 95.

7) What is the meaning of SSA SSA governments in the line number 141.

8) In this section (3.2. Malaria prevention and treatment) author must be write about the toxicity of these drugs especially liver stage drug and their effect in pregnant women.

9) The references is old and author must be incorporate new references in the revised manuscript.

10) There are some of punctuation and typographical errors throughout in the manuscript. Please correct it

Author Response

The manuscript entitled "An overview of malaria transmission mechanisms, control, and modeling". Title, abstract and overall rationale of work is well written and property explain. However, there are still some major concerns, which needs to be addressed before publication..

  • Author must be include two more keywords such as Malaria, Mosquito or suitable keywords.

We thank the reviewer for his suggestion and included the suggested keyword (Line 19)

  • Author describe methodology and explain how they collected the articles from different sources. It is better a flowchart should be added to the article to show the clear and attractive methodology.

We include a figure and its caption (lines 73-77)

  • This is review article and author explain many things about the malaria transmission mechanisms, control, and modeling, unfortunately, author did not showing any figure related to mechanism, transmission control and other. I suggest author they must be incorporate the figure here.

As suggested by the reviewer, we included a synthetic figure displaying the complexity of malaria and accounting for the next comment regarding the quality of the provided Fig.2 (lines 138-152).

  • Figure 2 is not attractive and low resolution. I suggest author they should increase the resolution and modified this representation.

We improve the quality of Figure 2 (lines 138-152).

5) Line number 88-90: author describe about the Plasmodium parasite transmission and they cite the references (9-10). These references are too old and it is better to incorporate new reference (doi: 10.1016/j.jare.2020.02.016) here.

We thank the reviewer for sharing this groundbreaking research with us. This allows us to learn more about the mechanism triggering the host immune system response even though in the suggested paper a rat model and different parasites are used.  Interestingly, the authors of the suggested paper referred to  Frischknecht (10.1101/cshperspect.a025478) when describing the bloodstream phase of the parasite. These authors cited older papers since the focus was not on the description of the transmission. Fundamentally, the transmission model has not been disproved by newly published papers. Nonetheless, we mentioned the immune inform the reader that we did not cover the host immune system response (lines 312-314)

  • Before used abbreviation author must be write first time in the manuscript after that they used. For example Anopheles gambiae. See the line number 95.

We defined the abbreviation (line 100)

  • What is the meaning of SSA SSA governments in the line number 141.

SSA was defined earlier in the paper (line 23).

  • In this section (3.2. Malaria prevention and treatment) author must be write about the toxicity of these drugs especially liver stage drug and their effect in pregnant women.

We thank the reviewer for suggesting the inclusion of drug toxicity. Although the description of drug toxicity for pregnant women is a fascinating idea, it is very challenging to include that in the review without deviating from the scope of the review for two reasons. First, given that pregnant women belong to a vulnerable group. Thus, if we include this group, we should equally include other vulnerable groups including children, and persons with comorbidity. Second, we mentioned only a few medications used in chemoprevention.

  • The references is old and author must be incorporate new references in the revised manuscript.

We mentioned this as a limitation of the study (lines 312-318)

  • There are some of punctuation and typographical errors throughout in the manuscript. Please correct it

We reviewed it accordingly.

Reviewer 2 Report

This is an intresting article on profilling on malaira control, transmission rate program.   
A few  minor suggestions here.  

1. Figure 2 too small to be read. Pls make it larger.
2. The article is rather short, perhaps the author can consider discussing how technology is transforming the landscape as it poised to malaria elimination program.
This scope is seriously lackiong in this article.

                https://doi.org/10.1101/721076
    https://www.nature.com/articles/s41467-021-21110-w
    https://www.nature.com/articles/nm.3622
    https://doi.org/10.1002/mrm.28387
     https://www.nature.com/articles/s42003-020-01262-z

3. Please comment on the sampling size and how it affect the interpretation ?  

Author Response

This is an intresting article on profilling on malaira control, transmission rate program.   
A few minor suggestions here.  

1. Figure 2 too small to be read. Pls make it larger.

Figure 2 is revised accordingly and the change can be observed in lines 138-151

  1. The article is rather short, perhaps the author can consider discussing how technology is transforming the landscape as it poised to malaria elimination program.
    This scope is seriously lackiong in this article.

    https://doi.org/10.1101/721076
     https://www.nature.com/articles/s41467-021-21110-w
        https://www.nature.com/articles/nm.3622
        https://doi.org/10.1002/mrm.28387
         https://www.nature.com/articles/s42003-020-01262-z

We thank the reviewer for this brilliant suggestion. Precisely, there is significant progress made worldwide over the last half a century to eliminate malaria. This progress can be observed through the significant reduction of the prevalence achieved with a tremendous improvement in technology allowing a better diagnosis and treatment of malaria. Given the cross-cutting topic reviewed, it appears challenging to dive into this subject. Nonetheless, we provide the readers with this topic as a limitation for our review (lines 312-314).

  1. Please comment on the sampling size and how it affect the interpretation ?  

Defining a sample size prior to a study is one of the approaches used in biomedical and socio-behavioral sciences to avoid bias and misrepresentation. Its definition assumes a given distribution is prior knowledge (literature or personal experience) and an erroneous definition might lead to a not representative conclusion. As such, it might appear that its definition is relevant to any study. Nonetheless, a narrative review intends to give a summary of a body of documented evidence. To minimize the self-selection bias and provide the reader with a representative summary, we described the selection method of the relevant papers (lines 48-78). Nonetheless, we mention the self-selection bias as a limitation (lines 314-316).

Round 2

Reviewer 1 Report

I have completed my evaluation of your manuscript and I found authors have addressed all the concerns raised in the previous version of the manuscript and the quality has improved after incorporating required modifications. Therefore, the manuscript may be considered for publication in this Journal.